# TOD-Flow: Modeling the Structure of Task-Oriented Dialogues

**Sungryull Sohn**[1*]    **Yiwei Lyu**[2*]    **Anthony Zhe Liu**[2]    **Lajanugen Logeswaran**[1]
**Dong-Ki Kim**[1]    **Dongsub Shim**[1]    **Honglak Lee**[1,2]
[1]LG AI Research    [2]University of Michigan, Ann Arbor

## Abstract

Task-Oriented Dialogue (TOD) systems have become crucial components in interactive artificial intelligence applications. While recent advances have capitalized on pre-trained language models (PLMs), they exhibit limitations regarding transparency and controllability. To address these challenges, we propose a novel approach focusing on inferring the TOD-Flow graph from dialogue data annotated with dialog acts, uncovering the underlying task structure in the form of a graph. The inferred TOD-Flow graph can be easily integrated with any dialogue model to improve its prediction performance, transparency, and controllability. Our TOD-Flow graph learns what a model can, should, and should not predict, effectively reducing the search space and providing a rationale for the model's prediction. We show that the proposed TOD-Flow graph better resembles human-annotated graphs compared to prior approaches. Furthermore, when combined with several dialogue policies and end-to-end dialogue models, we demonstrate that our approach significantly improves dialog act classification and end-to-end response generation performance in the Multi-WOZ and SGD benchmarks. Code available at: https://github.com/srsohn/TOD-Flow

## 1 Introduction

Task-Oriented Dialogue (TOD) systems have attracted significant attention due to their potential applications in personal assistants, customer support, and other interactive systems that necessitate human-like conversation (Balaraman et al., 2021; Zhang et al., 2020). Many of the recent advances in TOD have heavily leaned on pre-trained language models (PLMs) (He et al., 2022b; Wu et al., 2020) that are first pre-trained on a large corpus of data in an unsupervised manner, and then either fine-tuned (He et al., 2022b; Chen et al., 2019;

Wang et al., 2020) or subjected to few-shot prompting (Hudeček and Dušek, 2023; Labruna et al., 2023) to adapt them to specific dialogue domains. While these approaches have yielded commendable performance, they have limitations. Few-shot prompted models have been challenged by issues of transparency, controllability, and adaptability to specific domains, especially when working with only a few examples. The lack of understanding of their decision-making processes and fine-grained control over their output is often inadequate, which can result in sub-optimal conversational experiences. On the other hand, fine-tuned models are confronted with their own unique challenges. While they offer improved performance by aligning the model with task-specific semantics, this approach typically requires large annotated datasets and resources which can be a limiting factor in practice. Furthermore, these models often lack transparency, making it challenging to understand the reasons behind their decisions or predictions.

Some prior works (Raghu et al., 2021; Laradji et al., 2023) introduced workflow-based dialog models to handle the challenges in existing TOD models. These methods aim to explicitly model the structure of dialog in a graph format. Grounding the dialog in the graph offers benefits in terms of 1) elucidating the reasoning of system's decisions in terms of the relationships (*i.e.*, transparency); 2) allow human manipulation of the dialogue model via graph modification (*i.e.*, controllability) without retraining the dialog model. However, real-world dialogues were often unstructured, making it non-trivial to be modeled as a workflow, and the necessity of manually designing the domain-specific workflow or its elements limits its practical applicability.

To tackle these challenges, we propose to learn the *subtask graph* (Sohn et al., 2018) from task-oriented dialog data. Intuitively, the subtask graph

---

[*]Equal Contribution

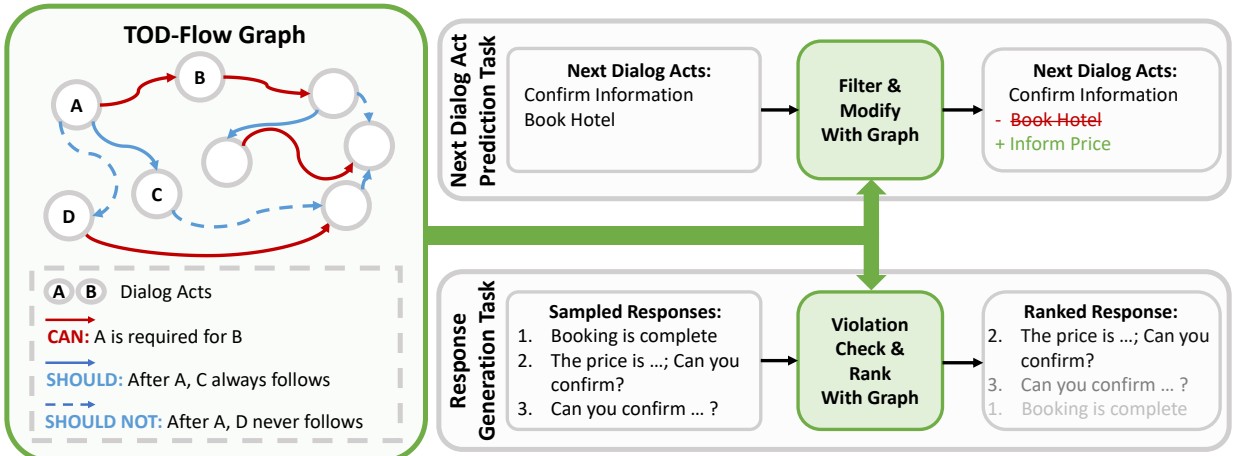

Figure 1: (Left) Our TOD-Flow graph captures the causal dependency between dialog acts in terms of can, should, and should not relationships. (Right) Intuitively, given the TOD-Flow graph, we can predict the relevant and irrelevant dialog act or responses based on the current dialog state. Based on the predicted relevance, we can filter and rank the base model outputs to enhance the prediction performance in both dialog act classification and end-to-end response generation tasks.

can predict the affordance (*i.e.*, availability) of the action from the status of environment and agent (*i.e.*, the progress of completing a task or the subtasks). The subtask graph framework has two major benefits: i) subtask graphs can be inferred from the demonstrations without any direct supervision (*e.g.*, video (Jang et al., 2023) or transcript (Logeswaran et al., 2023)), ii) subtask graphs can be combined with the base prediction model to improve its prediction since the subtask graph does not decide what to predict but instead suggests the affordable candidates of prediction.

**Contributions.** The main contribution of this work is generalizing the subtask graph framework into task-oriented dialog settings. To this end, we propose the *TOD-Flow graph*, which extends the subtask graph framework in three major aspects. First, we show that subtask graph can infer the relationship between dialog state and dialog acts without requiring any manual definition of nodes and edges in graphs. Second, in addition to the precondition (or *can* relationship), we present learning algorithms to model two novel relationships, *should* and *should not*, which provide more fine-grained control and improved prediction. For instance, a *can* relationship may represent that the system can make a payment only if the user confirms the payment. The *should* relationship may learn that if a user ask about the address of the hotel, the system should reply back. Conversely, a *should not* relationship may dictate that the system usually does not predict a farewell if the user's last utterance

implies a question. Third, we demonstrate that the inferred TOD-Flow graphs can enhance any dialog policy or end-to-end dialog system, whether fine-tuned or prompted, without the necessity of retraining.

## 2    Background

Our main contribution, the TOD-Flow graph, is an extension of the subtask graph framework (Sohn et al., 2018, 2020), which describes the causal dependency structure of a compositional task $\tau$ consisting of a set of subtasks. In the context of task-oriented dialogue, dialog acts can be seen as subtasks. Each subtask has a **precondition** that must be satisfied before the subtask can be performed. Note that the precondition is not the only relationship between subtask, and in Section 3.2 we extend it by incorporating other types of relationships. Since precondition describes the causal relationship between subtasks, it imposes a constraint on the order in which subtasks can be performed (*e.g.*, the system can make a payment *only after* the user confirms the payment). Formally, we define the precondition as a Boolean expression consisting of Boolean constants (*e.g.*, True or False), Boolean variables and logical connectives (*e.g.*, and ($\&$), or ($|$)). To illustrate, consider the precondition of subtask C: $f_C = \&(A, B)$, where the subtasks A and B must be completed before C is completed. It can be equivalently viewed as a Boolean function where inputs are Boolean variables indicating whether subtasks A and B are completed, and the output

represents whether the precondition $f_C$ is satisfied: $f_C(A = \text{True}, B = \text{False}) = \text{True} \& \text{False} = \text{False}$. Also, the boolean expression $f_C = \&(A, B)$ can be viewed as a **graph** with vertices consisting of subtasks and logical operators $V = \{A, B, C, \&\}$ and edges $E = \{A \rightarrow \&, B \rightarrow \&, \& \rightarrow C\}$ that represent preconditions. We will use these different views of the precondition (*i.e.*, as a boolean expression, graph or function) interchangeably. The **subtask graph** visualizes the preconditions $f_1, \dots$ of the subtasks (see Figure 1 for examples). We note that the subtask graph has been adopted in various settings (Liu et al., 2022; Sohn et al., 2020, 2022) and subsumes other task graph formats (Andreas et al., 2017; Boutilier et al., 1995; Sakaguchi et al., 2021), flowchart (Raghu et al., 2021), and workflow (Laradji et al., 2023).

## 3 TOD-Flow Graph Learning

### 3.1 Problem Formulation

For dialogue turn $t$, let $u_t$ be the user input and $r_t$ be the corresponding system response. The user inputs and system responses are represented as a set of *dialog acts* that labels the raw language utterance at each turn according to its category of meaning: $u_t, r_t \subset \mathcal{A}$, where $\mathcal{A}$ is the set of dialog acts. Let $d_t$ be the database query result that can be obtained through querying the database. Then, the dialog data $\mathcal{D}$ is a set of dialog trajectories $\mathcal{D}^\tau = \{(u_0, r_0, d_0, u_1, \dots), \dots\}$. Given the dialog data $\mathcal{D}^\tau$, the goal is to generate the *TOD-Flow graph $G$* that models the dependency between system acts, user acts, and database results in graph format.

**Challenges.** There are two main challenges in tackling this task. First, the information in the dialogue is noisy due to annotation errors such as missing or ambiguous dialog acts, slots, and values. Second, these dialog annotations only provide partial information about the underlying relationships between subtasks. Thus, we need to infer whether each relationship is satisfied or not from the dialogue annotations. We describe how we overcome these challenges in Section 3.3.

### 3.2 TOD-Flow Graph

For each dialog act $\mathbf{a}$, the TOD-Flow graph is defined in terms of three conditions: $\text{Can}_\mathbf{a}$, $\text{Shd}_\mathbf{a}$, and $\text{Shdnt}_\mathbf{a}$. Intuitively, $\text{Can}_\mathbf{a}$, $\text{Shd}_\mathbf{a}$ and $\text{Shdnt}_\mathbf{a}$ condition respectively defines whether the dialog act $\mathbf{a}$

| $a[n]$ \ Cond | Executed $(a[n] = 1)$ | Not executed $(a[n] = 0)$ |
|---|---|---|
| $f_n^{\text{Shd}} = 1$ | True positive | False positive |
| $f_n^{\text{Shd}} = 0$ | - | - |
| $f_n^{\text{Shdnt}} = 1$ | False positive | True positive |
| $f_n^{\text{Shdnt}} = 0$ | - | - |
| $f_n^{\text{Can}} = 1$ | True positive | - |
| $f_n^{\text{Can}} = 0$ | False negative | - |
| $f_n^{\text{BC}} = 1$ | True positive | False positive |
| $f_n^{\text{BC}} = 0$ | False negative | True negative |

Table 1: Confusion matrix of the TOD-Flow graphs (Shd, Shdnt, Can) and the BC baseline with respect to the dialog act label $a[n]$. The empty cell $(-)$ indicates that the label for each relationship is unavailable.

*can*, *should*, and *should not* be performed by the agent (user or system) at a given status. Similar to the precondition in subtask graph framework (see Section 2), each condition is defined as a Boolean expression. Also, it can be equivalently viewed as a Boolean function $f^{\text{Can}}, f^{\text{Shd}}, f^{\text{Shdnt}} : \mathbf{c} \mapsto \{0, 1\}$ or a graph (see Section 2), where $\mathbf{c} \in \{0, 1\}^{N_\tau}$ is the subtask completion (or dialog state) vector indicating whether $n^{\text{th}}$ subtask has been achieved (*i.e.*, $c[n] = 1$) or not (*i.e.*, $c[n] = 0$).

### 3.3 Learning TOD-Flow Graphs

**Dataset.** Given the dialogue data $\mathcal{D} = \{(u_t, d_t, r_t)\}$, we aim to build the graph inference dataset $\mathcal{D}_G = \{(\mathbf{c}_t, \mathbf{a}_t)\}$, from which we can infer the TOD-Flow graph $f^{\text{Can}}$, $f^{\text{Shd}}$, and $f^{\text{Shdnt}}$. The action set $\mathbf{a}_t$ is the set of dialog acts that were performed at turn $t$: $\mathbf{a}_t = u_t \cup r_t$. The completion set $\mathbf{c}_t$ is the set of dialog acts and database query that has ever been performed before turn $t$: $\mathbf{c}_t = \mathbf{c}_{t-1} \cup \mathbf{a}_{t-1} \cup \{d_{t-1}\}$.

**Shd inference.** When should condition is satisfied (*i.e.*, $f_n^{\text{Shd}}(\mathbf{c})=1$), the agent is required to perform the $n^{\text{th}}$ dialog act (*i.e.*, $a[n] = 1$). When the should condition is not satisfied (*i.e.*, $f_n^{\text{Shd}}(\mathbf{c})=0$), the should relationship has no effect on the policy. This relationship can be represented as the confusion matrix shown in the first two rows in Table 1. Accordingly, we maximize the true positive, while minimizing the false positive by maximizing the objective $J_{\text{Shd}}$ in Equation (1).

$$J_{\text{Shd}} = \mathbb{E}_{(\mathbf{c}, a[n])} \left[ \mathbb{I}(a[n] = 1 | f_n^{\text{Shd}}(\mathbf{c}) = 1) \right] \quad (1)$$

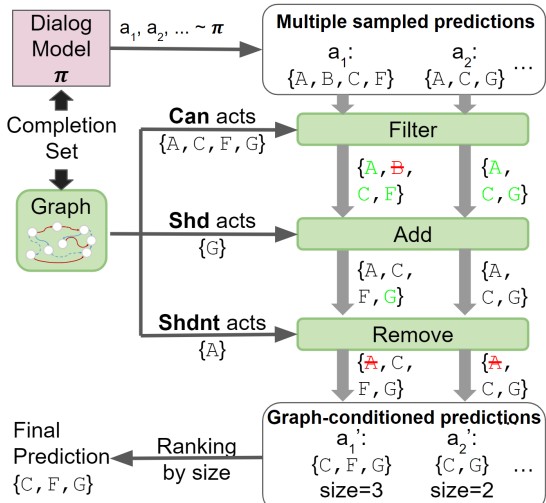

Figure 2: An illustration of our graph-conditioned dialog policy. We sample multiple predictions from base dialog policy $\pi$, then use our graph to filter, add and remove acts from each prediction following Algorithm 1. Then we obtain final prediction by selecting the graph-conditioned prediction with the largest size.

**Shdnt inference.** Similar to Shd, when the shoud-not condition is satisfied, (i.e., $f_n^{\mathsf{Shdnt}}(\mathbf{c}) = 1$), the agent is required *not* to perform the $n^{\text{th}}$ dialog act (i.e., $a[n] = 0$), and agent has a freedom to execute the $n^{\text{th}}$ dialog act when the condition is not satisfied (i.e., $f_n^{\mathsf{Shdnt}}(\mathbf{c}) = 0$). Table 1 summarizes the relationship between Shdnt and $a[n]$. We learn $f_n^{\mathsf{Shdnt}}$ by maximizing the following objective:

$$J_{\mathsf{Shdnt}} = \mathbb{E}_{(\mathbf{c},a[n])} \left[ \mathbb{I}(a[n] = 0 | f_n^{\mathsf{Shdnt}}(\mathbf{c}) = 1) \right]. \tag{2}$$

**Can inference.** By definition of Can (or precondition), a dialog act can only be performed ($\mathbf{a}_n = 1$) if its Can condition is satisfied (i.e., $f_n^{\mathsf{Can}}(\mathbf{c}) = 1$): a true positive case in Table 1. On the contrary, it is a contradiction if a dialog act $\mathbf{a}_n$ is performed while its Can is not satisfied (i.e., $f_n^{\mathsf{Can}}(\mathbf{c}) = 0$): a false negative case in Table 1. Thus, the $f^{\mathsf{Can}}$ can be learned by maximizing the following objective:

$$J_{\mathsf{Can}} = \mathbb{E}_{(\mathbf{c},a[n])} \left[ \mathbb{I}[f_n^{\mathsf{Can}}(\mathbf{c}) = 1 | a[n] = 1] \right]. \tag{3}$$

However, different from Shd and Shdnt, inferring Can is nontrivial, because if we maximize $J_{\mathsf{Can}}$, we get the trivial precondition: always true (i.e., $f_n^{\mathsf{Can}}(\mathbf{c}) = 1$ for all $\mathbf{c}$). Previous works handled this issue by either making additional assumptions (Hayes and Scassellati, 2016; Huang et al., 2019) or applying regularization (Jang et al., 2023).

**Algorithm 1** TOD-flow Graph-conditioned Dialogue Model

**Require:** Dialogue model $\pi$, TOD-flow graph $f^{\mathsf{Can}}, f^{\mathsf{Shd}}, f^{\mathsf{Shdnt}}$, Completion $\mathbf{c}$
**Ensure:** Sampled dialog acts $\mathbf{a}$
1: $\mathbf{a} \sim \pi$ ▷ Sample dialog acts from $\pi$
2: $\mathbf{a} \leftarrow \mathbf{a} \cup \{\mathbf{a}' | f_{\mathbf{a}'}^{\mathsf{Shd}}(\mathbf{c}) = 1\}$ ▷ Apply Shd
3: $\mathbf{a} \leftarrow \mathbf{a} \cap \{\mathbf{a}' | f_{\mathbf{a}'}^{\mathsf{Can} \wedge \neg \mathsf{Shdnt}}(\mathbf{c}) = 1\}$
4: ▷ Apply Can $\wedge$ ¬Shdnt
5: **return** $\mathbf{a}$

However, these approaches unavoidably introduces noise in learning, and require careful hyperparameter tuning to balance between objective and regularization. Instead, inspired by the fact that Can and ¬Shdnt (i.e., negation of Shdnt) applies to the policy in the same manner (i.e., the dialog act $\mathbf{a}$ can be performed if both $f_n^{\mathsf{Can}} = 1$ and $f_n^{\neg \mathsf{Shdnt}} = 1$), we propose to infer Can and Shdnt simultaneously as follows:

$$\begin{aligned} J_{\mathsf{Can} \wedge \neg \mathsf{Shdnt}} &= \mathbb{E}_{(\mathbf{c},a[n])} \big[ \mathbb{I}[f_n^{\mathsf{Can} \wedge \neg \mathsf{Shdnt}}(\mathbf{c}) = 1 | a[n] = 1] \\ &\quad + \alpha \, \mathbb{I}[f_n^{\mathsf{Can} \wedge \neg \mathsf{Shdnt}}(\mathbf{c}) = 0 | a[n] = 0] \big], \end{aligned} \tag{4}$$

where $\alpha$ determines the relative weight between Can and Shdnt in optimization. Intuitively, the agent can perform $n^{\text{th}}$ dialog act only if Can is satisfied and Shdnt is not satisfied.

**Baseline.** As an ablation model, we consider the behavioral cloning (BC) (Michie et al., 1990) objective, which tries to mimic the demonstration behavior:

$$J_{\mathsf{BC}} = \mathbb{E}_{(\mathbf{c},a[n])} \left[ \mathbb{I} \left[ f_n^{\mathsf{BC}}(\mathbf{c}) = a[n] \right] \right] \tag{5}$$

The confusion matrix is shown at the bottom of Table 1.

We can use any binary classification models to optimize the objectives (1), (4), and (5). We used the decision tree models in the experiment following the previous works (Boutilier et al., 1995; Huang et al., 2019; Sohn et al., 2020).

## 4 Graph-conditioned Dialog Modeling

We describe how the inferred TOD-flow graph $G$ can enhance the prediction performance of any off-the-shelf dialogue policies and end-to-end dialog systems.

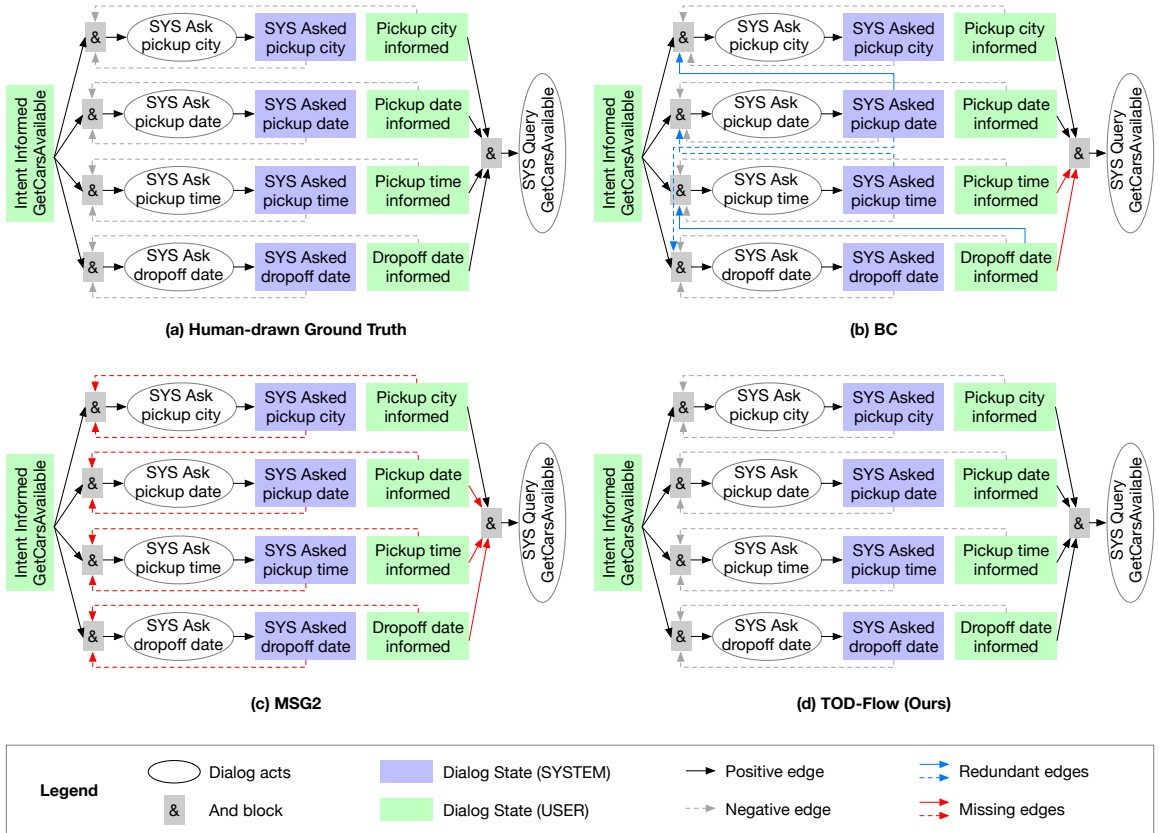

Figure 3: Comparing baselines and our method against human-drawn ground truth graph on a subpart of the `RentalCars_1` domain of SGD. Dotted lines are negative edges (logical negations). Red edges are missing edges and blue edges are redundant edges compared to the ground truth. In this scenario, the system is allowed to ask for a slot if user has informed the intent of finding an available car and has not yet informed this slot and the system have not yet asked for the slot; and the system can only query for a car when all 4 slots are informed. Our method perfectly matches the ground truth, while both baselines have incorrect edges (**BC** has 4 redundant edges and 2 missing edges, while **MSG**$^2$ has 11 missing edges).

## 4.1 Graph-conditioned Dialog Policy

The inferred TOD-flow graph (Shd and Can $\wedge$ $\neg$Shdnt) can propose the dialog acts that can, should, and should not be performed given the current dialog history (or completion set). Figure 2 and Algorithm 1 describes the entire process. Given a base dialogue policy $\pi^{\text{DP}}$, we sample the system acts from the policy $a \sim \pi^{\text{DP}}$, and filter, add and remove acts from the sampled system acts according to the Can, Shd, and Shdnt, respectively.

We can further improve the prediction performance if our baseline dialog model can be sampled multiple times with different results, as illustrated in Figure 2. We use the graph to condition each candidate result, then select the best one using a selection method such as most number of actions in set, candidate with least graph violations, etc. We empirically found that simply choosing the result with the most actions works best.

## 4.2 Graph-conditioned End-to-end Response Generation

End-to-end dialogue system directly reads and outputs the utterances in natural language form. Given the base end-to-end model $\pi^{\text{e2e}}$, we sample multiple system utterances from the base model (mostly via beam-search alternates, see Appendix A.2.1 for details). Then, we use a few-shot prompted `GPT-3.5-turbo` model to annotate each generated candidate utterance with the dialog act (see Appendix A.2.3 for details). Finally, we use the inferred graph to choose the best utterance that has the least *violation rate* (*i.e.*, portion of dialog acts that violates the inferred Can, Shd, and Shdnt conditions). Note that with this approach, all final responses still comes from the base end-to-end model, so the improvement is still upper-bounded by the capabilities of the model. Our graph simply presents a better candidate selection method.

| Models | SGD (24 domains) | | MultiWOZ (14 domains) | |
|---|---|---|---|---|
| | FLAN-T5 | GPT-turbo | FLAN-T5 | GPT-turbo |
| No Graph | 49.9% | 78.8% | 21.6% | 40.8% |
| **+BC** | 57.2% | 71.8% | 23.5% | 38.2% |
| **+MSG**[2] (Jang et al., 2023) | 52.4% | 79.4% | 23.8% | 40.2% |
| **+TOD-Flow** (ours) | **83.1%** | **89.2%** | **35.0%** | **48.2%** |

Table 2: Average F-1 scores of next system action prediction experiment on two datasets (SGD and MultiWOZ) and two large language models (FLAN-T5 and GPT-turbo) as few-shot predictors. We can see that while **BC** often damages performance, **TOD-Flow** consistently improves performance by a significant amount.

## 5 Experiments

We perform experiments to show that (1) the TOD-flow graph can be accurately predicted without any supervision, (2) our graph can improve the accuracy of dialog policy models, and (3) our graph can improve the quality of response generation in end-to-end dialog models.

### 5.1 Dataset

We used two standard TOD benchmarks. Schema-Guided Dialogue (SGD) (Rastogi et al., 2020) has over 20k task-oriented simulated conversations based on human-designed schema. SGD covers a wide range of domains (*i.e.*, different dialog acts and goals). We use 24 domains in SGD, and did not use the schema for experiment. Multi-WOZ (Budzianowski et al., 2020) has 10k human-human conversations on 14 domains. Since Mul-tiWOZ is collected from human-human conversations, the utterances are much more linguistically diverse than SGD. Also, different from SGD where the annotations are generated from the schema, annotations in MultiWOZ are labeled by human. Therefore, the annotations in MultiWOZ are often noisy (*i.e.*, inconsistent, wrong, or missing), which present additional challenge compared to SGD.

For both datasets, we obtain train/test splits of the dialogs within each domain (see Appendix A.1.1 for details). The training set is used for 1) inferring **TOD-Flow** graph, 2) building demonstration for few-shot prompted models, and 3) finetuning the finetuning-based models. The test set is only used for evaluation. For graph inference, we map the dialog act of user, database, and system to completion $\mathbf{c}$ and dialog act $\mathbf{a}$ vectors as described in Section 3.3.

### 5.2 Baselines

We compare three graph inference algorithms:

- **BC** learns to imitate the demonstration via behavioral cloning (see Section 3.3)

- **MSG**[2] (Jang et al., 2023) learns the sub-task graph by optimizing the $J_{\mathsf{Can}}$ (see Equation (3)) with complexity regularization.

- **TOD-Flow** (ours) is our **TOD-Flow** graph learning algorithm.

For fair comparison, we used the scikit-learn decision tree model (Pedregosa et al., 2011) for all the graph inference algorithms.

### 5.3 TOD-Flow Graph Inference

We first qualitatively compare the inferred graphs with the human-drawn graphs on RentalCars_1 domain in SGD dataset. We found that in general **TOD-Flow** produces graphs that agree with the human-drawn graphs much more often compared to baselines (**BC** and **MSG**[2]). Figure 3 illustrates the subpart of the inferred and human-drawn graph, where **TOD-Flow** inferred the graph perfectly matches the human-drawn graph, while the baselines missed important information (such as not requiring all 4 required slots to be informed before performing the query).

### 5.4 Task 1: Dialog Policy Learning

**Base Models.** We use two instruction-tuned large language models (LLM) as baseline dialog policy: FLAN-T5-xxl (Chung et al., 2022) and GPT-turbo[1]. At each turn, we prompt the LLM with five demonstration dialogues from train split of the same domain followed by the dialogue history, and ask the model to predict next system dialog acts. See Appendix A.1.2 for more details on prompting the LLM.

---

[1] https://platform.openai.com/docs/models/gpt-3-5

| | Can-Shdnt graph | Shd graph | Ranking method | SGD (24 domains) | | MultiWOZ (14 domains) | |
|---|---|---|---|---|---|---|---|
| | | | | FLAN-T5 | GPT-turbo | FLAN-T5 | GPT-turbo |
| No graph | ✗ | ✗ | Greedy | 49.9% | 78.8% | 21.6% | 40.8% |
| Graph ablations | ✗ | ✓ | Compliance | 70.4% | 82.5% | 26.1% | 41.6% |
| | ✓ | ✗ | Compliance | 65.6% | 83.5% | 30.4% | 44.5% |
| Sampling & ranking ablations | ✓ | ✓ | Greedy | 80.9% | 88.5% | 26.2% | 45.5% |
| | ✓ | ✓ | Majority | 75.2% | 88.1% | 20.7% | 45.0% |
| | ✓ | ✓ | Violation | 76.2% | 88.3% | 23.7% | 44.7% |
| | ✓ | ✓ | Uniform | 75.8% | 88.0% | 23.2% | 45.2% |
| Ours | ✓ | ✓ | Compliance | **83.1%** | **89.2%** | **35.0%** | **48.2%** |

Table 3: Ablation studies on graphs and ranking methods for next system action prediction. The numbers shown are average F-1 scores. We see that both Can-Shdnt and Shd graphs have significant contributions towards the performance and our Compliance ranking method outperforms greedy sampling or other ranking methods.

**Evaluation Protocol.** We sample 10 candidate predictions from the base models, which is filtered and ranked based on the graph (see Section 4.1) to choose the best prediction.

**Metric.** We measure F-1 score between the ground-truth and predicted system dialog acts at each turn and average over entire domains.

**Results.** Table 2 summarizes the F-1 score of each model on SGD and MultiWOZ. Overall, we observe that **TOD-Flow** consistently improves the prediction accuracy with a significant margin compared to other baselines **BC** and **MSG**[2] on all base models and all dataset. We also found that the improvements are bigger on FLAN-T5 compared to GPT-turbo. This indicates that the GPT-turbo already models the Can, Shd, and Shdnt to some extent, so that augmenting it with the graph provide less benefits. Since **BC** learns to mimic the exact behavior in demonstration, **BC** tends to dictate the base policy more aggressively and hurts the performance when combined with strong base model GPT-turbo. **MSG**[2] correctly models the precondition of dialog acts, but provides less benefit compared to **TOD-Flow** due to the conservative graph learning (i.e., complexity regularization) and lacking the ability to model Shd and Shdnt relations.

**Ablations.** To further justify our design choices, we performed ablation studies on two key components of **TOD-Flow**: the graphs and the ranking method after graph-conditioning. For filtering graphs, we examined the effect of Can-Shdnt and Shd graphs. Regarding the ranking method, we

compare the proposed ranking approach (i.e., Compliance) against various alternatives:

- **Greedy** ranks by likelihood of base LLM.
- **Compliance** ranks predictions by larger number of actions complying with the graph (i.e. rank by size in Figure 2).
- **Majority** chooses the majority prediction among the multi-sampled predictions.
- **Violation** ranks predictions by least number of actions filtered, added, and removed in graph conditioning.
- **Uniform** randomly chooses one of the multi-sampled predictions.

The results are shown in Table 3, and **TOD-Flow** outperformed all ablations, showing the necessity of all graphs and ranking by largest set.

### 5.5 Task 2: End-to-end Response Generation

**Base Models.** We use the three SOTA end-to-end dialogue models finetuned on MultiWOZ: GALAXY (He et al., 2022b), HDNO (Wang et al., 2020), and HDSA (Chen et al., 2019) as base models. ' Note that for GALAXY, since we were unable to reproduce the official prediction using the official repository, we report the result with both official prediction (GALAXY) and the greedy (i.e., beam search with beam width=1) prediction we obtained by running the official repository (GALAXY*).

**Evaluation protocol.** From each model, we first sample five system response utterances: one from official prediction[2] (Nekvinda and Dušek, 2021)

---

[2]https://github.com/Tomiinek/MultiWOZ_Evaluation/tree/master/predictions

| Graph | BLEU | Info | Succ | Score |
|---|---|---|---|---|
| HDSA | | | | |
| (no graph) | 20.74 | 87.20 | 78.00 | 103.34 |
| +BC | **20.76** | 86.80 | 77.70 | 103.01 |
| +MSG$^2$ | 20.71 | 86.90 | 77.90 | 103.11 |
| +TOD-Flow$^\dagger$ | 20.69 | 87.70 | 78.20 | 103.64 |
| +TOD-Flow | 20.51 | **88.10** | **79.00** | **104.06** |
| HDNO | | | | |
| (no graph) | 17.83 | 93.00 | 84.50 | 106.58 |
| +BC | 17.79 | 92.90 | 84.50 | 106.49 |
| +MSG$^2$ | 17.83 | 92.90 | 84.40 | 106.48 |
| +TOD-Flow$^\dagger$ | **18.08** | 93.10 | **85.10** | **107.18** |
| +TOD-Flow | 17.97 | **93.20** | 85.00 | 107.07 |
| GALAXY | | | | |
| (no graph) | **19.92** | 92.00 | 82.80 | 107.32 |
| +BC | 19.85 | 91.80 | 82.50 | 107.00 |
| +MSG$^2$ | 19.69 | 91.30 | 81.00 | 105.84 |
| +TOD-Flow$^\dagger$ | 19.86 | **92.40** | 83.30 | 107.71 |
| +TOD-Flow | 19.85 | **92.40** | **83.70** | **107.90** |
| GALAXY* | | | | |
| (no graph) | 18.88 | 90.70 | 80.70 | 104.58 |
| +BC | 18.88 | 90.20 | 80.40 | 104.18 |
| +MSG$^2$ | 18.70 | 89.80 | 80.50 | 103.85 |
| +TOD-Flow$^\dagger$ | 19.04 | **91.10** | 81.40 | 105.29 |
| +TOD-Flow | **19.12** | **91.10** | **82.30** | **105.82** |

Table 4: Results from the response generation experiment with Galaxy (He et al., 2022b), HDNO (Wang et al., 2020) and HDSA (Chen et al., 2019) as base models. The *Score* metric is computed by *Score* =*BLEU* +(*Info* +*Succ*)/2. **TOD-Flow**$^\dagger$ stands for the ablation model that excludes Shd graph from our **TOD-Flow**. As we can see, our graphs consistently improves all metrics except *BLEU* for all models, and we improve the official scores by a significant amount for each model.

and four from the model downloaded from the official implementation (see appendix A.2.1 for details). The graph conditioning process follows Section 4.2. As an ablation, we also evaluated our method without conditioning on Shd graphs.

**Metric.** We follow the standard evaluation metric using the official code (Nekvinda and Dušek, 2021), which computes three metrics on the MultiWOZ test set: *BLEU* (average *BLEU* (Papineni et al., 2002) scores between generated and ground truth response), *Info* (percentage of dialogs where the system presents an appropriate entity) and

*Succ* (percentage of dialogs where the task goals are achieved). The combined score is computed as *Score* =*BLEU* + (*Info* + *Succ*)/2. See Appendix A.2.2 for details on computing these metrics. We report all four metrics of the compared methods.

**Result.** We show the results in Table 4. We found that **TOD-Flow** can consistently improve *Info* and *Succ* metrics for all the base end-to-end dialog models. *BLEU* score fluctuates because it depends a lot on the exact wording of each response, which our graphs have no control over. The combined score consistently improves by 0.72, 0.49, 0.58, and 1.24 for HDSA, HDNO, GALAXY, and GALAXY*, respectively by conditioning with our **TOD-Flow**. Note that these score improvements are actually quite significant, as the difference between top and second top SOTA methods (GALAXY and HDNO) is only 0.74.

Next, we compare different graph generation methods: **BC**, **MSG**$^2$, and **TOD-Flow**. Overall, **TOD-Flow** consistently outperforms other baselines, **MSG**$^2$ and **BC**. In fact, the **MSG**$^2$ and **BC** often underperforms the greedy prediction without graph conditioning (*i.e.*, (no graph). Since the sampled predictions are in general much worse than the greedy prediction, unless the graph-based ranking is highly accurate, it often samples the prediction that is worse than greedy prediction. On the contrary, **TOD-Flow** is often able to accurately pick out the non-greedy better alternative to outperform the greedy predictions. We show one such example in Table 5. We also found that excluding Shd graphs (*i.e.*, **TOD-Flow**$^\dagger$) significantly reduced the improvement in all base models except HDNO. This shows that, while using only Can-Shdnt graphs also consistently improves performance of all base models, including Shd graphs can generally get even better results.

## 6 Related Work

**Task-oriented Dialog Systems.** There are two main classes of task-oriented dialog systems: *pipeline* systems and *end-to-end* systems. In *pipeline* approaches, the dialog system is segmented into various modules such as natural language understanding (NLU), dialog state tracking (DST), dialog policy and natural language generation (NLG) (Zhang et al., 2020). The NLU module first converts user language input into standardized dialog acts, slots and values. The DST module

| History | ... (A chat between USER and SYSTEM) _USER: No, I'm not ready to book it yet. Thank you!_ | |
|---|---|---|
| | **Utterance** | **Actions** |
| **GALAXY** | _SYSTEM: The address is [value_address], postcode [value_postcode], the phone number is [value_phone]_ | SYSTEM inform addr 
 SYSTEM inform post 
 SYSTEM inform phone |
| **GALAXY + TOD-Flow** | _SYSTEM: Is there anything else I can help you with?_ | SYSTEM reqmore |
| **Ground truth** | _SYSTEM: Just let us know when you are ready and we will be happy to assist you. Have a great day_ | SYSTEM reqmore 
 SYSTEM offer booking |

Table 5: Qualitative comparison of generated responses of **GALAXY** and **GALAXY+TOD-Flow** on the fourth turn of dialogue SNG0772 in MultiWOZ. **GALAXY**'s greedy-sampled response informs information such as phone number and postcode, which is out of context. Instead, our **TOD-Flow** chooses an alternate response that does not violate Can (_i.e._, SYSTEM inform phone) and complies with Shd (_i.e._, SYSTEM reqmore), and it turns out to be better and closer to the ground truth.

keeps track of the current dialog state in terms of the dialog acts, slots and values. Based on the current dialog state, the dialog policy module predicts the next action. Previous works have viewed the dialog policy task as a Markov Decision Process (MDP) (Kwan et al., 2023). Common approaches include reinforcement learning methods such as Q-learning, policy gradient (Lipton et al., 2017; Zhou et al., 2017; Gordon-Hall et al., 2020) with experience-replay (Malviya et al., 2022), or model-based planning (Peng et al., 2018). Lastly, given the current dialog states and the predicted dialog acts, the NLG module generates a response in natural language. We integrate the **TOD-Flow** graph into the dialogue policy module to improve its action prediction performance by learning what a model can, should and should not predict.

On the other hand, _end-to-end systems_ integrate all functionalities into one module. Many end-to-end models employ a singular language model to execute all four steps (He et al., 2022a). Alternatively, other methods bypass certain steps, generating the final response directly (He et al., 2022b; Wang et al., 2022). Pipeline systems are more interpretable and modular, allowing independent updates for enhanced control. Ours work uses the TOD-Flow graph to enhance end-to-end model responses by selecting the best one based on alignment with learned Can, Shd and Shdnt conditions.

**Graph-based Dialog Systems.** There has also been previous works attempting to integrate graphs into task-oriented dialog systems. Most of them focused on using graphs to represent or select information from a knowledge base (Yang et al., 2020,

2021). TGDM (Choi et al., 2016) attempted to create a dialog policy through manually constucted graphs, while a followup work (Kwon et al., 2018) proposed a rule-based system for automatically inferring these graphs given a working dialog policy. Raghu et al. (2021) tackles the task-oriented dialogue problem where the system must ground dialog utterances to the manually-defined flowcharts describing the procedure and adapt to unseen ones during testing. Laradji et al. (2023) aims to discover the workflow, a sequence of dialog acts with their respective slot values, from unseen conversation. Our **TOD-Flow** graph is constructed from labeled dialogue data without any human supervision. And our graph models the relationship between dialog acts and slots; _e.g._, what dialog act or updates in slot values can happen or not.

# 7 Conclusion

This work introduced a novel framework for improving the efficiency and predictive accuracy of task-oriented dialogues models. By leveraging the concept of subtask graph and generalizing it to a _TOD-flow graph_, we accurately inferred the latent task structure within a dialogue. As showcased through extensive experimentation with two public TOD datasets, the proposed technique has been proven to effectively generate accurate and human-interpretable graphs. Importantly, we have integrated these inferred graphs with a range of dialogue models, without necessitating retraining, resulting in a substantial enhancement in performance in both dialog act classification and end-to-end response generation.

## Limitations

Although our method can be directly used when there are multiple domains involved in a single dialog (by treating combination of domains as a single separate domain and creating graphs using dialogs that has the same combination of domains, similar to what we did for MultiWOZ), this approach is limited in that (1) we need to infer a graph for every combination of domains that is present (such as the multi-domain dialogs in SGD, where there are 24 different single-domains alone), and (2) we cannot easily generalize to unseen domain combinations (even if we have graphs for each individual domain). In the future, we would like to explore ways to directly combine graphs for individual domains into multi-domain graphs and thus address the two limitations above.

We also relied on action annotations from the datasets to infer graphs, which limits the applicability of our approach. It would be interesting to extend our approach to unannotated raw dialogues.

## Acknowledgments

This work was supported in part by grants from LG AI Research.

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

## A Experiment Details

### A.1 Next Action Prediction

#### A.1.1 Dataset Preprocessing details

In the next action prediction experiment, we are using 2 datasets: MultiWOZ (Budzianowski et al., 2020) and SGD (Rastogi et al., 2020). We first split each dataset into domains and train/test splits. For MultiWOZ, the dataset has an official splitting of train/val/test splits, so we follow the same splits; while MultiWOZ contains dialogs from 7 domains, 2 of them (police, hospital) have no test dialogs, thus we split MultiWOZ dialogs into 14 domains, including 5 single-domains (i.e. dialogs that only involves one of the 5 domains) and 9 multi-domains (i.e. dialogs that involves multiple domains, such as Hotel+Train). For SGD, since the official train/val/test splits often involves test schemas that does not exist in the train set, we decided to create our own train/test splits from the official training set. There are 24 different schemas that have single-schema dialogs in the official training set, so we treat each of these schemas as a separate domain and randomly split dialogs within each domain into train/test splits at a 9:1 ratio.

We then turn each dialog into a trajectory (as defined in section 3.1). Below is how we define the dialog actions within each domain of each dataset:

For SGD, since there are already very comprehensive dialog acts and slot annotations, we directly use the acts defined in the dataset (with a few re-naming) plus the slot annotations to build our set of all possible actions. In addition, since SGD provides explicit annotations about system's database queries, whenever the system queries database, we add an additional turn in our trajectory with one action "SYSTEM query <Intent>" and the status update would be either query success or query failure.

For MultiWOZ, we mostly also directly use the acts and slots in the dataset annotation to build our set of all possible actions, but we did some re-naming and re-organization to remove some redundant combinations of acts and slots (for example, "SYSTEM Booking-Inform <slot>" was changed into "SYSTEM OfferBook + SYSTEM inform <slot>"). Since there are no explicit annotation about database querying, we simply assume that the system looks up information before each "book/nobook" operation and add a corresponding status update to the utterance before these actions. We do not explicitly add "query" actions or additional turns to the trajectories.

Then, within each domain, we will use the trajectories of the train split dialogs to obtain graphs and also act as demonstrations for LLMs, and use the graphs to improve next action predictions on the test dialogs.

#### A.1.2 Large Language Model Prompting Details

In this experiment, we used GPT-turbo-3.5 as well as FLAN-T5 (Chung et al., 2022) as baseline next action predictors. We prompt the two LLMs using the exact same prompting method. For each domain, we first randomly select 10 dialogs from the training split, and then use their trajectories (i.e. actions and statuses of each turn) as demonstrations. Then for every system utterance in the test-split dialogs in the domain, we include as many of the demonstration trajectories as possible without exceeding the max token limit of the LLMs, and then we include the partial trajectory of the test dialog up to the turn where the next system actions needs to be predicted. Lastly, we ask the LLM to predict the next system actions, and we programmatically parse the results into individual action items. See Figure 4 for example prompt for SGD and MultiWOZ dataset.

When obtaining the baseline result for each model, we use the top prediction by probability by setting temperature to zero (thus the language model's generation is deterministic). When we need to multiple predictions for graph filtering, for FLAM-T5-xxl we simply do a beam-search of size 10; for GPT-turbo-3.5 we set temperature to 1 and sample 10 times.

### A.2 Response Generation

#### A.2.1 Getting Alternates from each model

For Galaxy (He et al., 2022b), we used the top choice of beam-size-1 as well as the top 3 choices of beam-size-5 as alternates. The overall ranking

of the 5 choices from highest to lowest are baseline, beam-size-1, beam-size-5 top choice, beam-size-5 second choice, beam-size-5 third choice.

For HDNO (Wang et al., 2020), since the official prediction baseline is the top choice of beam-size-5, we use the two choices from beam-size-2 and the second and third choice from beam-size-5 as alternates. The overall ranking of the 5 choices from highest to lowest are baseline (i.e. beam-size-5 top choice), beam-size-2 top choice, beam-size-5 second choice, beam-size-2 second choice, beam-size-5 third choice.

For HDSA (Chen et al., 2019) the process is slightly different. HDSA model consists of 2 parts: the first part (predictor) predicts the actions the system will perform (although in a very different format than what we do in our next action prediction experiment, so not directly comparable), and the second part (generator) uses the output of the predictor to generate the response. If we fix the predictor output and do beam-search on the generator only, the actions within the generated response will almost always be identical, which renders our method useless. Therefore, we created our alternates by tweaking a hyperparameter in the predictor a little bit. The output of the predictor is a binary vector, and the post-sigmoid logits of the predictor is converted to the binary vector by a threshold. The HDSA official code repository has 0.4 as the default threshold, and we changed the threshold around that value and used the different generated vectors as inputs to the generator to obtain our alternates. The overall ranking of the 5 choices from highest to lowest are baseline, threshold-0.4, threshold-0.375, threshold-0.35, threshold-0.325.

### A.2.2 Evaluation Details

We use the official MultiWOZ_Evaluation repository to evaluate the BLEU/INFORM/SUCCESS metrics. Since our policy-learning setting assumes that we have access to the ground truth dialog state before the utterance, we use the ground truth dialog state and active domains in the evaluation scripts (by removing dialog state / active domain predictions and only including the response in the prediction file). This is necessary because we found that active domain predictions affect the INFORM/SUCCESS metrics, and incorrect active domain can increase/decrease INFORM/SUCCESS randomly. Therefore, to ensure fairness and consistency, we always use the ground truth active domain during evaluation.

### A.2.3 Using GPT as NLU unit

We prompt GPT-turbo-3.5 to convert the candidate responses into action sets. For responses for dialogs in each domain, We first provide randomly selected dialogs from the training split of the same domain together with their ground truth system actions as demonstrations. Then, we specify the desired output format and provide an example of the output format. Lastly, we provide the dialog history of the current candidate responses, and ask GPT to give us actions to all candidate responses (i.e. the baseline + 4 sampled from the models). We start with 6 demonstrations, and we reduce the number of demonstrations by one iteratively if the total number of tokens exceeds the maximum token limit of the model (4097). We show one example prompt for responses from Galaxy (He et al., 2022b) together with the GPT completion in Figure 5.

We evaluated the quality of this NLU process by using this process to predict actions of the ground truth responses and compare the predicted actions to the ground truth actions on a subset of the test dialogs. We found that on average our NLU's predicted actions achieves an average F-1 score of 77.6%, which is okay but far from perfect, and the imperfectness of our NLU brings additional challenge to our task.

## B   Human-drawn GT graphs for SGD

In order to perform qualitative assessment of the quality of our graphs, we manually drew graphs for 10 domains in SGD. We show 2 such graphs in Figure 6 for RideSharing_1 schema and Figure 7 for RentalCars_1 schema respectively.

```
Demonstration:
USER Actions: Inform number of days, Inform Intent ReserveHotel
Status: Informed number of days, Intent Informed ReserveHotel
SYSTEM Actions: Request hotel name, Request check in date
Status: Requested hotel name, Requested check in date
USER Actions: Inform number of rooms, Inform hotel name, Inform check in date
Status: Informed number of rooms, Informed hotel name, Informed check in date
SYSTEM Actions: Confirm hotel name, Confirm check in date, Confirm number of days, Confirm number of rooms
Status: Confirmed hotel name, Confirmed check in date, Confirmed number of days, Confirmed number of rooms
USER Actions: Request phone number, Request destination, Affirm
Status: Requested phone number, Requested destination, Affirmed
SYSTEM Actions: Query ReserveHotel
Status: DATABASE N>0 ReserveHotel
SYSTEM Actions: Inform phone number, Inform destination, Notify Success
Status: Informed phone number, Informed destination, Notified Success
USER Actions: Thank You, Goodbye
Status: Thanked, GOODBYE
SYSTEM Actions: Goodbye
Status: END OF DIALOG

Demonstration:
[... omitting 9 demonstrations …]

Based on the demonstrations above, predict the next line of Action and Status below:

USER Actions: Inform Intent ReserveHotel
Status: Intent Informed ReserveHotel
SYSTEM Actions: Request hotel name, Request check in date
```

```
Demonstration:
USER act: hotel-inform type, hotel-inform area, hotel-inform stars
Status: hotel-inform type, hotel-inform area, hotel-inform stars
SYSTEM act: hotel-inform name
Status: hotel-inform name
USER act: hotel-inform price
Status: hotel-inform price
SYSTEM act: booking-offerbook, hotel-inform price, hotel-inform stars
Status: booking-offerbook, hotel-inform price, hotel-inform stars
USER act: general-thank
Status: general-thank
SYSTEM act: general-bye
Status: general-bye

Demonstration:
[... omitting 5 demonstrations …]

Based on the demonstrations above, predict the next line of SYSTEM Actions below:

USER act: hotel-inform area
Status: hotel-inform area
SYSTEM act: hotel-request type, hotel-request price
```

Figure 4: Example of using GPT-turbo-3.5 to do next system action prediction from the Hotels_1 domain of SGD (top) and from the Hotel+Train domain of MultiWOZ (bottom). We first provide a few demonstration trajectories in the prompts, then we ask the large language model to complete the prompt. The text highlighted in green is the GPT completion, while everything before that is our input prompt. We give the exact same prompt to the other LLM (FLAN-T5-xxl).

Demonstration:
USER Utterance: i am planning a trip to [value_departure] and looking for a train to get me there . can you help me find 1 please ?
SYSTEM Utterance: we have many trains to cambridge , where are you departing from ?
SYSTEM Actions: SYSTEM train-request depart | SYSTEM train-inform num choices | SYSTEM train-inform dest
USER Utterance: i am departing from [value_departure] and i would like to go to [value_destination] . i would like my train to arrive by [value_arrive] and leave on [value_day] .
SYSTEM Utterance: the [value_id] arrives in [value_destination] by [value_arrive] , can i book a ticket for you ?
SYSTEM Actions: SYSTEM train-inform arrive | SYSTEM train-inform dest | SYSTEM train-inform id | SYSTEM train-offerbook
[... omitting 7 turns of  interactions …]
USER Utterance: not at this time . thank you for all the information . i am all set . goodbye .
SYSTEM Utterance: have a nice day , goodbye
SYSTEM Actions: SYSTEM general-bye

Demonstration:

[... omitting 5 more full dialog demonstrations  …]

 Now, we are going to ask you to predict the actions from several candidate responses. Your answer format must fit the following example:

(1) [value_name] is a [value_food] restaurant .
(2) there are [value_choice] restaurant-s that meet your criteria . 1 serves [value_food] food and the other serves [value_food] food . do you have a preference ?
(3) there are [value_choice] restaurant-s that meet that criteria . 1 serves [value_food] food and the other serves [value_food] food . which would you prefer ?

Your output should be:

(1) SYSTEM restaurant-inform name | SYSTEM restaurant-inform food
(2) SYSTEM restaurant-ask-for-selection food | SYSTEM restaurant-inform num-choices | SYSTEM restaurant-inform food
(3) SYSTEM restaurant-ask-for-selection food | SYSTEM restaurant-inform num-choices | SYSTEM restaurant-inform food

Now consider the following partial dialog:

USER Utterance: i am looking for a train that leaves on [value_day] and arrives by 10.30

Based on the demonstrations above, predict the SYSTEM Actions for each of the following candidate SYSTEM response:
(1) i can help you with that . where are you departing from and arriving ?
(2) where are you departing from and arriving to ?
(3) there are [value_choice] trains that meet that criteria . where are you departing from and arriving to ?
(4) there are [value_choice] trains available . where are you departing from and arriving ?
(5) there are [value_choice] trains available . where are you departing from and arriving to ?

Your Answer:
(1) SYSTEM train-request depart | SYSTEM train-request dest
(2) SYSTEM train-request depart | SYSTEM train-request dest
(3) SYSTEM train-inform num choices | SYSTEM train-request depart | SYSTEM train-request dest
(4) SYSTEM train-inform num choices | SYSTEM train-request depart | SYSTEM train-request dest
(5) SYSTEM train-inform num choices | SYSTEM train-request depart | SYSTEM train-request dest

Figure 5: Example of using GPT-turbo-3.5 to obtain actions from candidate responses from Galaxy (He et al., 2022b). The parts highlighted in green are completion from GPT, and everything before that is our prompt.

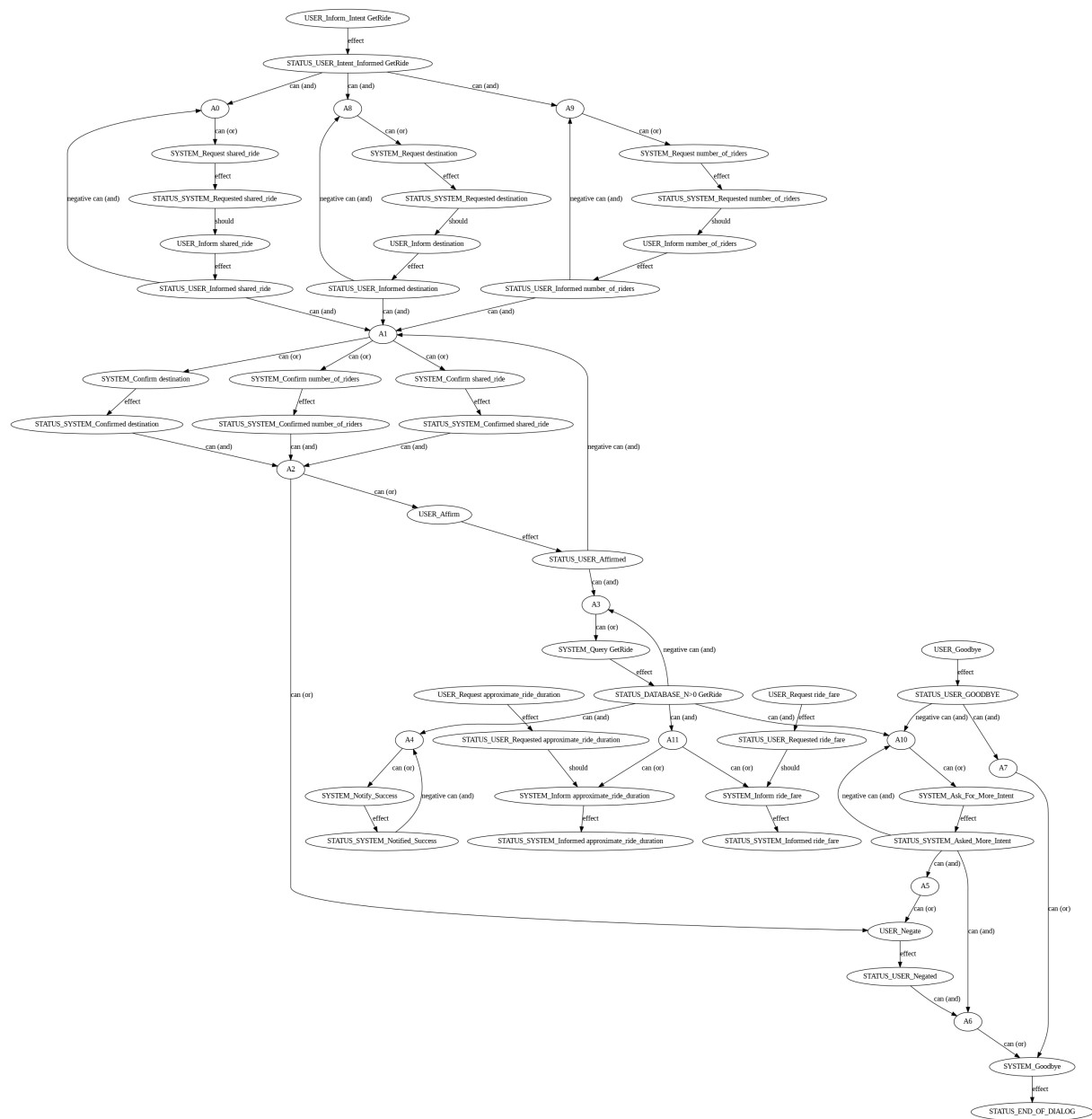

Figure 6: Human-drawn ground truth graph for RideSharing_1 domain in SGD.

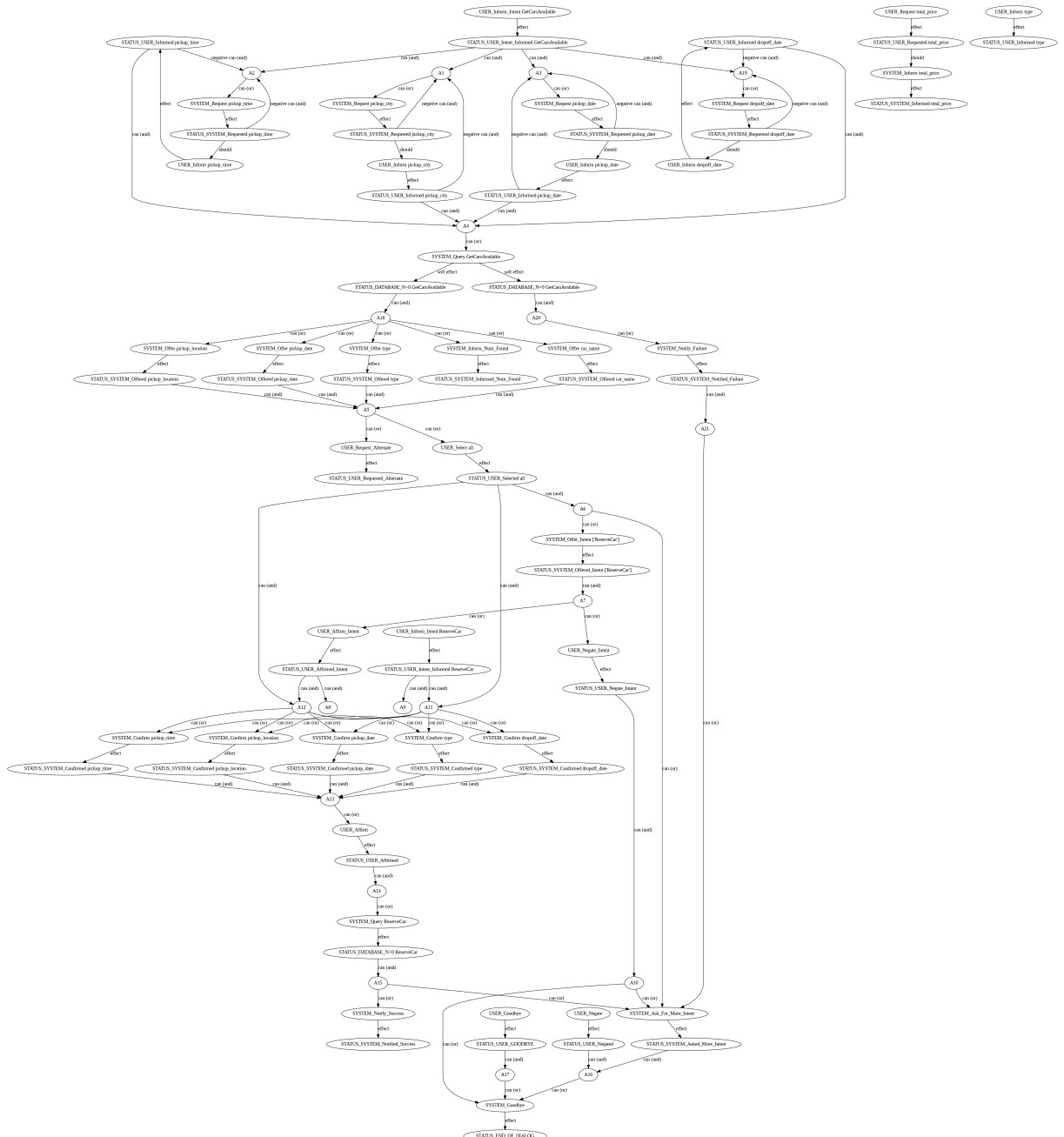

Figure 7: Human-drawn ground truth graph for RentalCars_1 domain in SGD.