# OpenReview forum: "TOD-Flow: Modeling the Structure of Task-Oriented Dialogues"
_EMNLP/2023/Conference — EMNLP 2023 Main_

### Official Review · Reviewer_uYLX · 2023-08-01

**Soundness:** 3

**Excitement:**

4: Strong: This paper deepens the understanding of some phenomenon or lowers the barriers to an existing research direction.

**Paper Topic And Main Contributions:**

The paper proposes a novel approach to address the limitations of pre-trained language models (PLMs) in Task-Oriented Dialogue (TOD) systems regarding transparency and controllability. The approach focuses on inferring the TOD-flow graph from dialogue data annotated with dialogue acts, revealing the underlying task structure in a graph format. The inferred TOD-flow graph can be integrated with any dialogue model to enhance prediction performance, transparency, and controllability. The graph guides the model on what to predict, what not to predict, and provides a rationale for the model's predictions. The experiments demonstrate that the proposed TOD-flow graph closely resembles human-annotated graphs and significantly improves dialog act classification and end-to-end response generation performance in benchmark datasets.

Strengths:

1.The paper introduces a novel method for inferring the TOD-flow graph from annotated dialogue data, which enhances transparency and controllability in task-oriented dialogue systems.
2. The paper includes a rich set of experiments, demonstrating the effectiveness of the proposed approach in improving dialog act classification and end-to-end response generation performance.

Weaknesses:

Some aspects of the proposed model are not thoroughly described, leaving room for ambiguity.

**Questions For The Authors:**

In addition, I have several questions for the authors:

1.In the 3.1 Problem Formulation section, the authors state that the goal of the TOD-flow graph learning task is to generate the TOD-flow graph G given dialog data D. Could the authors elaborate on how the model is trained to generate the TOD-flow graph? What is the structure of the model, and what is the format of the dataset used for training? Providing a concrete example would be helpful to understand the process better.

2.In the "Graph-conditioned End-to-end Response Generation" section, the authors use the GPT-turbo model to annotate each generated candidate utterance with the dialogue act. How does the model ensure the accuracy of GPT-turbo's generated annotations? Additionally, how does the inferred graph guide the selection of the best response?

**Reasons To Accept:**

1.The paper introduces a novel method for inferring the TOD-flow graph from annotated dialogue data, which enhances transparency and controllability in task-oriented dialogue systems.
2. The paper includes a rich set of experiments, demonstrating the effectiveness of the proposed approach in improving dialog act classification and end-to-end response generation performance.

**Reasons To Reject:**

Some aspects of the proposed model are not thoroughly described, leaving room for ambiguity.

**Reproducibility:**

3: Could reproduce the results with some difficulty. The settings of parameters are underspecified or subjectively determined; the training/evaluation data are not widely available.

**Reviewer Confidence:**

3: Pretty sure, but there's a chance I missed something. Although I have a good feel for this area in general, I did not carefully check the paper's details, e.g., the math, experimental design, or novelty.

---

> ### Author Rebuttal · Authors · 2023-08-29
>
> We thank the reviewer for the helpful comments. We are glad to see that the reviewer acknowledges that 1) our method is novel and exciting, and 2) the paper presents a rich set of experiments demonstrating the effectiveness of the method. We address reviewer concerns below.
>
> > Detailed process of graph inference: data and model
>
> Let $D_{G}=$ {($c_t, a_t$)} be the graph inference dataset, introduced in the first paragraph of Section 3.3. The action set $a_t$ is the set of dialog acts that were performed at turn $t$. The completion set $c_t$ is the set of dialog acts and database query that has ever been performed before turn $t$. Then, we construct the TOD-flow graph in terms of **Can**+**Shdnt** and **Shd** conditions from the dataset $D_{G}$. Intuitively, the goal is to find the binary function $f: c \mapsto$ \{0, 1\} that maximizes the objectives in equation (1) and (4) for **Shd** and **Can**+**Shdnt**, respectively. We model the Boolean function with a binary decision tree, and use the Classification and regression tree (CART) algorithm to fit the binary decision tree to the dataset $D_{G}$. At each branching step of precondition inference for $n$-th action, the algorithm chooses the optimal feature (or each component of completion set $c_t$) that best discriminates the positive and negative samples (i.e., $a_t[n]=0$ or $1$) in terms of gini impurity [1]. The resulting decision tree (i.e., Boolean function) can be in turn represented as either Boolean expression or TOD-flow graph as explained in Section 2. We will include this detail in the revised paper.
>
> [1] Breiman, Leo; Friedman, J. H.; Olshen, R. A.; Stone, C. J., 1984, Classification and regression trees. Monterey, CA: Wadsworth & Brooks/Cole Advanced Books & Software
>
> > Accuracy of NLU module (i.e., GPT-turbo’s generated annotations) in end-to-end experiment
>
> We reported the accuracy of GPT-turbo-based annotations in Appendix A.2.3. Our few-shot prompted GPT-turbo scored 77.6% accuracy on predicting the dialog act annotation in MultiWOZ, which is reasonable but still far from perfect. We believe that with a more sophisticated NLU module with higher annotation accuracy, our method will generate more accurate graph and provide bigger improvement on base models.
>
> > How does the inferred graph guide the selection of the best response?
>
> Given the multiple responses sampled from the base end-to-end dialogue model, our graph can be used to select the best response among them. The process is described in Section 4.2, lines 309-312. Intuitively, we measure how many *violations* were made in each response with respect to the **Can**, **Shd**, and **Shdnt** conditions in the inferred graph. Specifically, given a response, we use a few-shot prompted GPT annotator to annotate the response with dialog acts and slots. Then, we examine the predicted annotation with respect to the **Can**, **Shd**, and **Shdnt** conditions in the inferred graph. For example, if the response contains a (dialog act, slot) pair whose **Can** condition is not satisfied, then it is a violation of **Can** condition. We choose the response with minimum violation rate (i.e., among all the predicted annotations, how many annotations violate the graph). We will include this detailed explanation in the revised paper.

---

### Official Review · Reviewer_CAsX · 2023-08-04

**Soundness:** 4

**Excitement:**

3: Ambivalent: It has merits (e.g., it reports state-of-the-art results, the idea is nice), but there are key weaknesses (e.g., it describes incremental work), and it can significantly benefit from another round of revision. However, I won't object to accepting it if my co-reviewers champion it.

**Paper Topic And Main Contributions:**

This study presents a new framework to enhance the efficiency and predictive accuracy of task-oriented dialogue models. By using the idea of a subtask graph and expanding it to a TOD-flow graph, the hidden task structure within a dialogue can be accurately determined. Extensive experiments with two public TOD datasets have demonstrated the effectiveness of this method in generating accurate and easily understandable graphs. Crucially, these inferred graphs have been integrated with various dialogue models without the need for retraining, leading to significant improvements in both dialogue act classification and end-to-end response generation.

**Reasons To Accept:**

- The propose method show its effectiveness in graph structure inference as well as two subtasks relevant to dialogues.
- The writing of this paper is clear and the figure is illustrating.

**Reasons To Reject:**

- This paper is not well motivated. Although the performance improved, it is not clear to me why using such structure how it is different from previous works.
- Lack of comparison and baselines with works that focus on dialogue structures such as [1][2]
- Although the efficiency is not the most focus of this paper, i believe it should be at least discussed.


[1] Towards Efficient Dialogue Pre-training with Transferable and Interpretable Latent Structure
[2] DialogVED: A Pre-trained Latent Variable Encoder-Decoder Model for Dialog Response Generation

**Reproducibility:**

4: Could mostly reproduce the results, but there may be some variation because of sample variance or minor variations in their interpretation of the protocol or method.

**Reviewer Confidence:**

3: Pretty sure, but there's a chance I missed something. Although I have a good feel for this area in general, I did not carefully check the paper's details, e.g., the math, experimental design, or novelty.

---

> ### Author Rebuttal · Authors · 2023-08-29
>
> We thank the reviewer for the helpful comments. We are glad to see that the reviewer acknowledges that 1) our experiment shows the effectiveness of our graph inference method on two dialog tasks, 2) our work provides sufficient support for our main claim, and 3) the paper is well-written. We address reviewer concerns below.
>
> > Reasons for using the graph structure beyond performance gains
>
> Representing real-world tasks (e.g. videos, dialogs) as complex structures offers multiple benefits. First, it improves model transparency, as the graph explains predictions by showing actions the model can, should, and shouldn't take. Second, it enables controllability by modifying the graph to change model behavior. For example, a banking chatbot's loan criteria could be updated by editing the graph, without retraining the entire model. Third, the representation enhances accessibility of information to humans (Vicol et al. 2018; Wang and Gupta 2018; Zhao et al. 2022). Fourth, it provides downstream advantages like aiding planning and execution (Liu et al. 2022; Sohn et al. 2020).
>
> [1] Vicol, P.; Tapaswi, M.; Castrejo ́n, L.; and Fidler, S. 2018. MovieGraphs: Towards Understanding Human-Centric Situations from Videos. In CVPR.
>
> [2] Wang, X.; and Gupta, A. 2018. Videos as Space-Time Region Graphs . In ECCV.
>
> [3] Zhao, B.; Li, H.; Lu, X.; and Li, X. 2022. Reconstructive Sequence- Graph Network for Video Summarization. TPAMI.
>
> [4] Liu, A. Z.; Sohn, S.; Qazwini, M.; and Lee, H. 2022. Learning Pa- rameterized Task Structure for Generalization to Unseen Entities. In AAAI.
>
> [5] Sohn, S.; Woo, H.; Choi, J.; and Lee, H. 2020. Meta Reinforcement Learning with Autonomous Inference of Subtask Dependencies. In ICLR.
>
> > Novelty: how is the method different from previous works
>
> Please refer to the first response to Reviewer irFm.
>
> > Comparison with learning latent dialogue structure [1, 2]
>
> We thank the reviewer for suggesting relevant works [1, 2]. We agree that [1, 2] and ours share the high-level motivation: capturing the structure of dialog. However, there are several significant differences between our direction and [1, 2]. Here are the summary of differences:
> * The notion of “structure” differs significantly: ours model the dependency (or **Can**, **Shd**, **Shdnt** conditions) between dialog acts and slots, while [1, 2] tries to model the context, progress, persona, etc
> * In turn, the learning methodology also differs: ours use an inductive logic programming method to capture the logic rule, while [1, 2] uses variational inference and self-supervised learning to model the latent representations.
> * Our graph is independently inferred from the dialogue data; this inference process is independent of the base dialogue model used. Then the inferred graph works as an add-on to any base models, without any training. In [1, 2], the latent structure is a part of dialogue model, and the entire model is pre-trained (and finetuned) on large data.
>
> > Analysis: Efficiency of each method
>
> We first measured the efficiency of each method in terms of the graph construction time (in seconds). Below is the graph construction time for the 5 domains (Attraction, Taxi, Train, Hotel, Restaurant) dialogs from MultiWOZ:
>
> | Domain | TOD-Flow No Shd | BC       | MSG2        |TOD-Flow    |
> |------------|--------------|-------------|-------------|-------------|
> | Attraction |  2.11 | 1.11 |  1.80 | 14.31 |
> | Taxi       |  2.65 | 2.82 | 1.76 | 28.96 |
> | Train      |  5.97 | 7.33 | 12.27 | 402.60 |
> | Hotel      |  30.72 | 26.26 | 49.72 |  540.16 |
> | Restaurant |  47.74 | 35.96 | 61.70 | 438.74 |
>
> Next, we measured the inference time for graph-based action ranking. For action ranking, it computes whether the **Can**, **Shd**, and **Shdnt** conditions are met for each action. Per each turn, all the graph inference methods took less than 0.002 second for evaluating the condition satisfaction, which is negligible compared to the response generation time: 0.46 sec for FLAN-T5 and 0.58 sec for GPT-turbo.

---

### Official Review · Reviewer_irFm · 2023-08-05

**Soundness:** 4

**Excitement:**

3: Ambivalent: It has merits (e.g., it reports state-of-the-art results, the idea is nice), but there are key weaknesses (e.g., it describes incremental work), and it can significantly benefit from another round of revision. However, I won't object to accepting it if my co-reviewers champion it.

**Paper Topic And Main Contributions:**

The paper introduces TOD-Flow, a novel approach that models the structure of task-oriented dialogues by inferring TOD-flow graphs from annotated dialogue data. The main contributions include the development of a method that uncovers the underlying task structure in the form of a graph, leading to better transparency, controllability, and prediction performance in dialogue models, as well as providing a rationale for the model's predictions by effectively reducing the search space.

**Questions For The Authors:**

1. Instead of inducting the graph automatically, are there any ways to manually design such a graph? If we use a finely curated graph, how would it affect the model's performance?

**Reasons To Accept:**

1. Augmenting the dialogue policy with a graph structure is well-motivated, and is well-backed by the explorations on other tasks (e.g., videos and transcripts) as discussed in the paper.

2. The task sub-graph is induced from raw dialogue data without introducing auxiliary annotation efforts. This promotes the availability of applying the proposed approach to more domains/scenarios.

**Reasons To Reject:**

1. It is unclear how much novelty the proposed approach offers in comparison to existing sub-graph applications such as those by Jang et al. and Logeswaran et al. It seems the proposed approach is extending a well-established technique by introducing some TOD-tailored rules (should & shouldn't inference). An in-depth discussion should be added.

2. The evaluation protocol for the experiments involving the policy w/ and w/o graph augmentation is confusing. An off-the-shelf model (either a trained T5 or ChatGPT) is used for evaluation. However, the base models have not been tuned using the graph-augmented TOD data. This raises doubts about whether considering graph structures during training could lead to performance gains.

3. The selection of baselines might not be comprehensive. In Table 2, only prompting methods are considered. It is crucial to consider policy models that learn through traditional BC or RL rather than solely relying on a pre-trained LLM.

**Reproducibility:**

3: Could reproduce the results with some difficulty. The settings of parameters are underspecified or subjectively determined; the training/evaluation data are not widely available.

**Reviewer Confidence:**

4: Quite sure. I tried to check the important points carefully. It's unlikely, though conceivable, that I missed something that should affect my ratings.

---

> ### Author Rebuttal · Authors · 2023-08-29
>
> We thank the reviewer for the helpful feedback. We are encouraged that the reviewer found our work well motivated and is widely applicable to more domains/scenarios. We address the reviewer’s concerns below.
>
> > Novelty of TOD-Flow compared to previous works [Jang et al., 2023, Logeswaran et al., 2023]
>
> We note that Logeswaran et al., 2023 used the graph inference algorithm proposed in MSG2 (Jang et al., 2023) as-is, so we will only compare with MSG2.
> 1. Our method infers **Can**, **Shd**, and **Shdnt** relations while MSG2 only infers the precondition which is equivalent to **Can** relation. Note that **Shd**, and **Shdnt** relations can also be inferred and used for the next subtask prediction task in MSG2 setting.
>
> 2. Ours and MSG2 optimize different objectives for precondition (or **Can**) inference. If we ignore minor implementation details, we can directly compare their objective functions as follows:
> $$
> \begin{align}
> J_\text{ours}   &= P(f_n(c)=1, a[n]=1) + \alpha P(f_n(c)=0, a[n]=0)\newline
> J_\text{MSG2}&= P(f_n(c)=1, a[n]=1) / P(f_n(c)=1)+ \beta C( f_n )\newline
> J_\text{ILP}     &= P(f_n(c)=1, a[n]=1),
> \end{align}
> $$
> where $f_n(\cdot)$ is the n-th precondition function, $c$ is the completion set, $a[n]$ is whether the n-th action (\ie, dialog act and slot) is performed, and $\alpha$ and $\beta$ are the hyperparameters.
> MSG2 and ours modify the original ILP objective ($J_\text{ILP}$) differently. At a high level, MSG2 prevents overly optimistic preconditions by multiplying $1/P(f(c)=1)$, and adds complexity regularization $C(f)$ to prevent overfitting. Ours instead infers **Can** and **Shdnt** relationships *simultaneously* (see Section 3.3), where **Shdnt** regularizes the **Can** inference. Our experiments show ours produces a significantly better graph than MSG2.
>
> 3. The implementations differ: ours learned a decision tree while MSG2 used beam search in the precondition space, as the complexity regularization term in MSG2 is hard to be incorporated into decision tree learning.
>
> > “The evaluation protocol using untuned base models on graph-augmented data raises doubts about potential gains from graph-informed training.”
>
> In our experiment, all the models only use the given annotated dialogue data, and our graph is not used to augment or alter the data. Also, all the models use the same un-tuned base models: FLAN-T5 and GPT-turbo. Once the graph is inferred from given annotations in the training set, it is then used to filter, add, and remove predictions (see Figure 2 in the paper) of a base model during testing. Thus, we claim that the reported performance gains purely come from conditioning the dialogue model with our inferred graph.
>
> > Comparison with traditional BC or RL-based dialogue policy baselines
>
> Our base dialogue policies (GPT-turbo and FLAN-T5) observe only five dialogue trajectories as a part of the prompt. We observed that the BC or RL dialogue policies trained on five trajectories severely overfit to the training trajectories. Thus, we did not consider BC or RL-based baselines in a low-data regime.
>
> > Ablation experiment with hand-drawn graph
>
> We manually curated **Can** and **Shdnt** conditions for the *RideSharing_1* domain in SGD by manually examining schemas and raw dialogs. We skipped the **Shd** condition due to its complexity. We then compared this to our inferred **Can** and **Shdnt** conditions by evaluating dialog policy. The results are summarized in the table below.
>
> |                       | FLAN-T5 | GPT-Turbo |
> |-----------------------|---------|-----------|
> | No Graph              | 0.574   | 0.800     |
> | Human-drawn           | 0.773   | 0.857     |
> | Ours (Can+Shdnt-only) | 0.780   | 0.864     |
> Table. Average F-1 scores of next system action prediction on *RideSharing_1* domain.
>
> Summary of observations:
> * The Human-drawn graph achieved similar improvements as ours (see Table above) in next system action prediction. Ours slightly outperforms the human-drawn graph.
> * Our inferred graph models more complex dialog relations than human-drawn graphs (226 vs 62 edges). Humans tend to simplify or miss minor relationships.
> * In general, Human-drawn graphs are costly and do not scale, taking our annotator 3 hours even for the simplest domain, *RideSharing_1*. Our automated inference is cheaper and more scalable, taking only around 2 seconds for graph inference.

---

### Meta-Review · Area_Chair_PxWM · 2023-09-19

**Recommendation:** 4

**Metareview:**

This paper presents TOD-flow, a new approach to extract conversation flow graphs from dialogue dataset annotated with dialogue acts. The paper is well-written with helpful illustrating figures. The propose method shows its effectiveness in graph structure inference as well as two subtasks relevant to dialogues. The selection of baseline in evaluation was challenged by reviewers -- which the authors agreed to and provided further comparison during rebuttal.

---

### Decision · Program_Chairs · 2023-10-07

**Decision:**

Accept-Main

**Comment:**

This paper presents TOD-flow, a new approach to extract conversation flow graphs from dialogue dataset annotated with dialogue acts. The paper is well-written with helpful illustrating figures. The propose method shows its effectiveness in graph structure inference as well as two subtasks relevant to dialogues. The selection of baseline in evaluation was challenged by reviewers -- which the authors agreed to and provided further comparison during rebuttal.